# Study of a Modified Time Hardening Model for the Creep Consolidation Effect of Asphalt Mixtures

**DOI:** 10.3390/ma15082710

**Published:** 2022-04-07

**Authors:** Yunming Ma, Hongchang Wang, Kang Zhao, Lizhu Yan, Dagang Yang

**Affiliations:** School of Civil Engineering, Nanjing Forestry University, Nanjing 210037, China; mym2257@126.com (Y.M.); zkas@njfu.edu.cn (K.Z.); 18851170185@163.com (L.Y.); ydg1215@163.com (D.Y.)

**Keywords:** modified time hardening model, modified Burgers model, time hardening model, single penetration creep, asphalt mixtures, viscoelastic surface fitting analysis

## Abstract

In the past, most researchers have explained the three-stage creep behavior of asphalt mixture in detail. Still, there is no reasonable model to describe the creep of the consolidation effect. To accurately describe the consolidation effect of an asphalt mixture during the viscoelastic deformation process, a modified time hardening model was established by using the Malthus model and the Logistic function to change its creep strain and creep compliance. According to the characteristics of asphalt mixture creep, a single penetration creep test was conducted for high-elasticity modified asphalt mixtures at different temperatures (20 °C, 40 °C, 60 °C) and various loading levels (0.55 MPa, 0.70 MPa, 0.85 MPa, 1.00 MPa). The test results showed that the effect of stress on deformation within the normal range of variation was more significant than that of temperature. In addition, the test results were simulated by the modified time hardening model using surface fitting and compared with a time hardening model and a modified Burgers model. A fitting analysis showed that the modified time hardening model more accurately represents the asphalt mixture’s consolidation effect and creep behavior. Therefore, the modified time hardening model can better show the consolidation effect in the creep process.

## 1. Introduction

Because asphalt mixtures have a good bearing capacity and anti-deformation ability, they are usually applied to the pavement surface. However, asphalt mixtures are viscoelastic materials, which are easily affected by temperature, stress, and their material properties [1,2,3,4,5,6]. Therefore, many flexible pavements’ distress, such as corrugation and ruts, are related to the viscoelastic creep behavior [7,8]. According to the available studies, there are three different forms of creep in materials in general. At low stress or low temperature, the creep behavior of the material exists in only two stages. Due to the hardening property [9], the strength of the material gradually increases, and the strain eventually approaches constant [10], which is the creep consolidation effect studied in this paper. At higher temperatures and stress, the typical three-stage creep behavior appears. The second stage disappears at very high pressure or temperature, and the first and third stages are adjacent [11].

The behavior of viscoelastic materials is between linear elasticity and ideal viscosity, and idealized models can describe their constitutive relationship. Many researchers have explored the irrecoverable deformation of pavement materials by rheological models of viscoelastic masses of asphalt mixtures. The common mechanical models include the Kelvin, Burgers, modified Burgers, Zener, and Delft-Xahu models. Chang et al. [12] compared the Kelvin model with the Burgers model and verified that the Burgers model has better utility and accuracy. The Burgers model connects the Maxwell and Kelvin model in series, which can express both the elastic deformation hysteresis and account for the simultaneous action of the mixture viscosity and elasticity. Li et al. [13] used the Burgers model to explain the creep behavior and viscoelastic principles of asphalt concrete based on the relationship between the steady-state creep rate and the experimental temperature and load stress. The viscoelastic deformation of asphalt mixtures does not increase indefinitely with the extension of the load time, but the increment decreases with time and eventually tends towards a stable value [14]. Compared with the traditional Burgers model, the modified Burgers model proposed by Xu et al. [15] can describe the consolidation effect generated by the mixture under pressure, and its accuracy is also higher than that of the Burgers model [1].

Bailey proposed an empirical model in the form of a power law for viscoelastic deformation [16]. According to subsequent studies, the Bailey–Norton model integrates the effects of time and stress on deformation, and it has two different manifestations: time hardening and strain-hardening [17,18,19,20]. According to American Association of State Highway and Transportation Officials (AASHTO 2015), the power function model always accurately predicts asphalt pavement deformation and is mainly used for the long-term prediction of pavement rutting [21]. These models are widely used to accurately describe the high-temperature creep of asphalt mixtures due to fewer parameter constraints and intuitive physical meaning [22,23]. However, this model can only characterize the primary and secondary stages of the asphalt mixture creep process. It cannot effectively describe the consolidation effect of the actual pavement under pressure.

In past studies, uniaxial or triaxial creep tests mainly tested the mixture [1,24,25]. The uniform force that characterizes this loading method on the asphalt mixtures and its creep parameters can be obtained [26]. A uniaxial penetration test can get the material’s shear strength, and the rutting can be predicted by exploring the creep law under repeated load [27]. Cárdenas et al. [28] used ANSYS (Finite Element Software) to simulate and calculate the size effect of cylindrical asphalt mixture samples, and the proposed modified penetration creep compliance matched the creep compliance of DSR (Dynamic Shear Rheometer). Fadil et al. [29] obtained the shear relaxation modulus of asphalt mixture by indentation testing, and the validity of the method was proved. In comparison, the loading method of a single penetration test ignores the effect of boundary conditions and is more consistent with the actual force conditions of the pavement.

This research is dedicated to proposing a new creep model, which can show the creep consolidation effect of materials. The time hardening model is modified based on logistic function, and the deformation and strain evolution in the static creep process can be simulated accurately. Single penetration creep tests were applied to test high-elasticity modified asphalt mixtures at different temperatures and stresses. Then, the modified Burgers model, time hardening model, and modified time hardening model were evaluated based on the test results.

## 2. Viscoelastic Model Analysis

Viscoelastic models can be generally divided into theoretical models of viscoelastic mechanical behavior and empirical models in power function form. This section reviews the models commonly used to describe asphalt mixtures. In addition, the Malthus model and Logistic model are used to modify the creep rate and creep compliance of the time hardening model.

### 2.1. Burgers Model

The Maxwell and Kelvin elements are connected in series to form a four-unit four-parameter model that can jointly describe the creep behavior of asphalt mixtures. The Burgers model with four parameters is expressed as:(1)$$\varepsilon =\sigma_{0}J=\sigma_{0}\left[\frac{1}{E_{1}}+\frac{t}{\eta_{1}}+\frac{1}{E_{2}}\left(1-e^{-\tau t}\right)\right]$$

As shown in Figure 1, where $\tau =\frac{E_{2}}{\eta_{2}}$. *ε* is the current strain, *σ*_0_ is a constant force, *J* is the creep compliance, *t* is the loading time, *E*_1_ is the elastic modulus of the Maxwell element, *η*_1_ is the viscosity coefficient of the Maxwell element, *E*_2_ is the elastic modulus of the Kelvin element, and *η*_2_ is the viscosity coefficient of the Kelvin element. From Equation (1), when *t→**∞*, *ε→**∞*, and the creep rate tend towards a constant value, the deformation rises linearly. However, in practical engineering, the deformation of asphalt concrete eventually appears to stabilize with time, and the rate gradually decreases. The fundamental reason is that the viscosity coefficient of the Maxwell element is constant, thus causing a permanent increase in deformation.

### 2.2. Modified Burgers Model

Wang et al. [30] argued that the viscous flow deformation of asphalt mixtures does not extend indefinitely with the time of load action. However, the increment of viscous deformation gradually decreases with increasing time, and the viscous deformation will eventually stabilize, which is the consolidation effect. To describe the creep of asphalt mixture better, the Burgers model is modified. The external dashpot in the Burgers model, which represents the flow deformation properties of the material, is extended to a generalized dashpot whose strain decreases with time and eventually stabilizes. If $\eta_{1}=Ae^{Bt}$, the model is as follows:(2)$$\varepsilon =\sigma_{0}J\left(t\right)=\sigma_{0}\left[\frac{1}{E_{1}}+\frac{1}{AB}\left(1-e^{-Bt}\right)+\frac{1}{E_{2}}\left(1-e^{-\tau t}\right)\right]$$
where $\tau =\frac{E_{2}}{\eta_{2}}$, *A* and *B* are parameters of *η*_1_. As shown in Figure 2, when *t**→**∞*, *ε* is a constant value (${\varepsilon =\sigma }_{0}(\frac{1}{E_{1}}+\frac{1}{AB}+\frac{1}{E_{2}})$) in stable loading, and the deformation rate gradually decreases. The Burgers model and the modified Burgers model are theoretical models which differ from power function models. They have more theoretical assumptions and a narrower range of constraints, so the fitting accuracy is not as good as in power function models [31].

### 2.3. Time Hardening Model

The time hardening model [19] is a common empirical model that is the most used one in finite element software to describe creep processes in viscoelastic materials. According to the Bailey–Norton creep law, the creep strain rate of a material is related to the loading stress, temperature, and time, and it is also often called the Bailey–Norton model [32]. When at a constant temperature, the model is:(3)$${\dot{\varepsilon }}_{cr}=C_{1}\sigma_{e}{}^{C_{2}}t^{C_{3}}$$
where ${\dot{\varepsilon }}_{cr}$
is the creep strain rate. $\sigma_{e}$ is the equivalent stress. *t* is the stress action time.
$C_{1}$, $C_{2}$, and $C_{3}$ are the parameters. The time variable of Equation (3) is integrated to obtain the time hardening model:(4)$$\overset{t}{\underset{0}{\int }}C_{1}\sigma_{e}{}^{C_{2}}t^{C_{3}}dt=\frac{C_{1}\sigma_{e}{}^{C_{2}}t^{C_{3}+1}}{C_{3}+1}$$

The creep compliance of the time hardening form can be obtained by definition as:(5)$$J=\frac{\varepsilon }{\sigma_{e}}=\frac{C_{1}\sigma_{e}{}^{C_{2}-1}t^{C_{3}+1}}{C_{3}+1}$$

Generally, $C_{1}{,\;C}_{2\;}\geq \;0$, $-{1<C}_{3\;}\leq \;0$. From Equations (3)–(5), it can be seen that the strain rate decreases with increasing time, which reflects the hardening behavior of the material, and the strain increases with increasing action time. In the time hardening model, the strain (creep compliance) is closely related to the action of stress and time. Similar to the Burgers model, the time hardening model does not represent the consolidation effect of the material.

### 2.4. Modified Time Hardening Model

The whole process of creep is affected by time *t*, stress *q*, and temperature *T*. Therefore, the relationship formula of the influence factor can be established, which is expressed by *X*, as shown in Equation (6):(6)X=f(T,q,t)

According to existing studies, the creep development of asphalt mixtures has three different strain stages [33], which are shown in Figure 3. In the tertiary stage, the creep strain varies exponentially with time in the form of rising [34]. The Malthus model is common in statistics and is used to represent the exponentially increasing trend in population size in an ideal state. The Malthus model represents the tertiary stage of asphalt mixture creep. Assume that the state of change of the three stages of creep of asphalt mixture is produced under the action of the influence factor *X*. With *r* being the growth rate (*r > 0*) and *ε* being the creep strain at the corresponding moment, the general form of the creep third tertiary process is Equation (7):(7)$$\frac{d\varepsilon }{dX}=r\varepsilon$$
which is solved to obtain:(8)$$\varepsilon =ae^{rX}$$
where *a* is the regression parameter. Equation (8) demonstrates a standard power function model. Different models can be derived due to the other creep influences studied. When only the effect of time is considered, Equation (8) will show that the strain increases exponentially under the action of time, which can indicate the tertiary stage. In addition, a can be expressed as the starting point of the damage stage (the dividing point of the second and third phases).

The following discussion will demonstrate that this model fits well with the empirical model.

Let $X=\frac{blnN}{r}$, the Monismith model [35] can be derived:
(9)$$\varepsilon_{\rho }=aN^{b}$$Let $X=\frac{blnN}{r}$
and a=ε(1), the Superpave model [36] can be derived:
(10)$$log\varepsilon_{\rho }=\log\varepsilon \left(1\right)+blogN\;$$Let $r=-{\left(\frac{\rho }{N}\right)}^{\beta }$, the Tseng and Lytton model [37] can be derived:
(11)$$\varepsilon_{\rho }=\varepsilon_{0}e^{-{\left(\frac{\rho }{N}\right)}^{\beta }}\;$$The computational model proposed by Zhou et al. [38] synthesizes the different strain rate characteristics in the three stages of creep. Its form is a sectional function. A power function is applied uniformly for the deceleration and accelerated damage stages to describe the relationship between permanent strain and loading repetition.

The deceleration stage:(12)$$\varepsilon_{\rho }=aN^{b}\;N<F_{s}\;$$

The accelerated damage stage:(13)$$\varepsilon_{\rho }=\varepsilon_{F_{N}}+d\left(e^{f\left(N-F_{N}\right)}-1\right)\;N\geq F_{N}\;$$
where *N* is the number of repeated loads, ε(1) is the permanent strain after the first loading cycle, a, b, β, ρ, $\varepsilon_{0}$ are all regression parameters. $F_{s}$ is the critical loading repetitions of the deceleration stage and the stationary stage, $F_{N}$ is the critical loading repetitions of the stationary and accelerated damage stages. Similarly, they are empirical models in the form of power functions. Therefore, using the Malthus function to express the creeping trend is reasonable. In addition, there are many forms of power function models. This paper will not discuss them and their influence factor *X* in detail.

Unlike the above empirical model, this paper aims to find the creeping trend that can describe a gradual decrease in strain and a rate that eventually goes to zero. A smooth curve must be used for simulation to ensure the continuity and accuracy of the whole creep process. Here, the creep compliance J(X) will be introduced for discussion, which is the same as the simulation process of Equation (7).
(14)$$\frac{dJ\left(X\right)}{dX}=rJ\left(X\right)\;$$

The solution to this Equation is known to be:(15)$$J\left(t\right)=ae^{rX}\;$$

In this connection with the time hardening model, two influencing factors of time and stress are introduced for analysis. Combining Equations (5) and (15), the expression of influence factor *X* is obtained as follows:(16)$$X=\frac{(C_{2}-1)ln\sigma +(C_{3}+1)lnt}{r}\;$$

In fact, the creep rate of viscoelastic materials gradually decays instead of increasing exponentially [39,40]. Here, a logistic model is introduced to express the consolidation effect of asphalt mixture creep. The logistic model originates from a developmental law of nature and is often used to model trends in the impact of the environment on organisms. As shown in Figure 4, the logistic model is constrained by the upper limit at the later stage. The rate of change is affected by a gradually decreasing trend, which precisely matches the creeping trend in the asphalt mixture when it is compressed. Therefore, the Malthus time hardening model is modified according to the logistic model to create a modified time hardening model. Suppose the growth rate is *k* and the deformation limit value is *D*. According to the integration of the mechanical model. This differential equation represents the creep compliance, and the influence factor is shown as Equation (17):(17)$$\frac{dJ\left(X\right)}{dX}=kJ\left(X\right)\left(1-\frac{J\left(X\right)}{D}\right)\;$$
where *k* and *D* are undetermined parameters. The solution of Equation (17) is known to be Equation (18):(18)$$J\left(X\right)=\frac{D}{1+ae^{-kX}}\;$$
where a is integration constant. This study is based on the Bailey–Norton law, which considers only the effects of two external factors: time and stress. Therefore, the general form is obtained by substituting Equation (16) into Equation (18), as shown in Equation (19):(19)$$J\left(\sigma ,t\right)=\frac{D}{1+a\sigma^{\frac{-k(C_{2}-1)}{r}}t^{\frac{-k\left(C_{3}+1\right)}{r}}}\;$$

Let $n=\frac{-k(C_{2}-1)}{r}$ and $m=\frac{-k(C_{3}+1)}{r}$. Then, the simplified expression can be written in the following form:(20)$$J\left(\sigma ,t\right)=\frac{D}{1+a\sigma^{n}t^{m}}\;$$

Then, the relationship for creep strain can also be derived as Equation (21):(21)$$\varepsilon \left(\sigma ,t\right)=\sigma J\left(\sigma ,t\right)=\frac{D\sigma }{1+a\sigma^{n}t^{m}}\;$$
where *D*, *a*, *n* (*n < 0*), and *m* (*m < 0*) are regression coefficients that depend on the creep properties of the asphalt mixture itself and the effect of temperature. *D* is also expressed as a coefficient of the deformation limit, which can intuitively express consolidation effects in the viscoelastic deformation of materials. The lower the *D* value is, the higher the rutting resistance of the asphalt mixture. From Equation (21), the following can be seen under the action of static loading:If t→∞, ε→ D·σ. The higher the stress is, the higher the deformation limit.If t=0, ε=0. Therefore, this is in line with reality.

## 3. Materials and Methods

High-elasticity asphalt mixtures are widely used in the pavement of steel bridge decks due to their better viscosity, low-temperature ductility, and elastic recovery rate. Moreover, these mixtures exhibit good high-temperature performance and rutting resistance.

### 3.1. Asphalt Binder

The high-elasticity asphalt used in this study was obtained from Japan’s Nichireki Corporation and tested following Chinese national standards (CNS) and AASHTO guidelines (1993). The results are shown in Table 1.

### 3.2. Aggregate

Aggregates #1~#4 are all Nanjing Jinshilei basalt rock material. The mineral powder is Nanjing Jiangning limestone ore powder. The natural sand is river sand from the Ganjiang River in Jiangxi Province. The synthetic gradation uses #4 material: natural sand is a selected gradation of 1:1, and natural sand is blended with a 12:7 ratio of medium-coarse sand and Guoyuan fine sand. According to CNS, the material is sieved, and then the mass of different sieves is weighed, and the different sieved materials are mixed.

### 3.3. Specimen Preparation

According to the Chinese Standard Test Method for Asphalt and Asphalt Mixtures for Highway Engineering (JTG E20-2011, 2011), the specimens of the high-elasticity modified asphalt mixture were formed by a Superpave gyratory compactor (SGC) with a vertical pressure of 600 KPa, a rotation angle of 1.25°, and a speed of 30.1 rpm. The dimensions of the specimens were 150 mm in diameter and 100 mm in height.

### 3.4. Test Method

In this paper, the optimal amount of asphalt was determined to be 6.0% using the Marshall design method, consistent with the design index of the high-elasticity modified asphalt mixture of 5.7% to 6.3%. According to AASHTO guidelines and the Chinese National Standard (JTG E20-2011, 2011), a 78-B7015 universal testing machine (UTM) manufactured by CONTROLS, Milan, Italy, was used in this study for the single penetration static creep test. The UTM records the force and displacement during the test to obtain the creep curve. The test specimens and the loading device were preheated for 5 h before the test to reach the target temperature. To better verify and support the constitutive model, the test environment temperature was set to 20 °C, 40 °C, and 60 °C, and static loadings of 0.55 MPa, 0.7 MPa, 0.85 MPa, and 1.0 MPa were carried out in these environments. To reduce the error, three sets of parallel tests were performed.

As shown in Figure 5, a 42 mm diameter indenter (JTG E20-2011) was placed in the center of the specimen to control the overall force on the specimen. A plastic film was set at the loading position, and the indenter was lubricated to prevent the loading device from sticking to the specimens. The specimens were pre-pressed for 5 min and 0.1 MPa before loading, followed by unloading for 30 min. The loaded samples were set for 60 min to ensure the adequacy of the asphalt mixture deformation. In this paper, only the deformation of the asphalt mixture during continuous loading is considered. This loading method provides a realistic circumferential pressure applied by the circumferential mix to the pressurized portion of the asphalt mixture [41].

## 4. Results and Discussion

### 4.1. Creep at the Same Temperature Levels

Figure 6 shows the strain trend in the material under different loads at the same temperature conditions. Under the influence of various stresses, the strain trends of the high-elasticity modified asphalt mixture were approximately the same at each temperature. However, with increasing stress, the relative growth level of the strain between the different stress levels also showed different trends, as shown in Table 2. The SROI represents the strain rate of increase in different stress (temperature) intervals, and the STD represents its corresponding standard deviation. It can be seen that the strain rate of increase in the asphalt mixture is more significant under a high-stress state. The strain rate of increase reaches a maximum (103.9%) in high-temperature and high-stress conditions (40 °C, 0.85 MPa~1.0 MPa), while the strain rate of increase reaches a minimum (10.9%) in the high-temperature and low-stress state (60 °C, 0.55 MPa~0.7 MPa). When the temperature is low, the stress dramatically affects the strain rate of increase. This is because the deformation of the mixture under low-temperature conditions is mainly based on the elastic effect. However, the strain rate of increase decreases at a high temperature due to the decrease in elastic effect and accumulation of viscous effect of asphalt mixture, which reduces the influence of stress on strain growth.

### 4.2. Creep at the Same Stress Levels

Figure 7 shows the strain trend in the asphalt mixtures under the same stress and different temperatures. At low temperatures, the strain tends to grow very slowly and decreases in the latter stages of stress loading. At the same time, the asphalt mixtures are gradually compacted, and the strength gradually increases. However, as the temperature increases, the strain rate of the asphalt mixture rises. Therefore, a more significant strain occurs. The high temperature enhances the viscous effect and accelerates its shear deformation [42]. Therefore, there is no continuous squeezing of the mixture. According to Table 3, the materials all produced a more pronounced trend in deformation growth at high temperatures. For example, the strain of the high-elasticity modified asphalt mixture reached 0.495% at 0.55 MPa and 60 °C, which is a 50.9% increase relative to the strain at 0.55 MPa and 20 °C.

According to the above analysis, under different conditions, the creeping trend presented by the high-elasticity asphalt mixture is the same, and the creep strain rate decreases with increasing time. Stress and temperature affect the asphalt mixture in the same trend but with different effects. For example, Table 2 and Table 3 show that the strain rate of increase is 99.4% when stress is increased from 0.55 to 1.00 MPa. On the other side, in Figure 7a, when the temperature is changed from 20 °C to 60 °C, the strain rate of increase is only 65.4%. So, the effect of stress is higher in the range of stress and strain considered in the test program. At low temperatures and low pressures, the strain rate of increase in the asphalt mixtures at late loading (3600 s) is prolonged. Predictably, when the loading time is long enough, the strain rate decreases to 0 when the deformation stabilizes, and viscoelastic deformation has reached a steady state.

## 5. Model Fitting Analysis and Verification

1stOpt software (5.0, 7D-Soft High Technology Inc., Beijing, China) was used to perform surface fitting on the time hardening model, the modified time hardening model, and the modified Burgers model. The way of surface fitting is shown in Figure 8, Figure 9 and Figure 10. The results are shown in Table 4, Table 5 and Table 6. All three models can simulate the creep process of high-elasticity asphalt mixtures. From the evaluation results (R^2^ and RMSE: Root Mean Squared Error) of the fitted parameters, the time hardening model and the modified time hardening model have similar accuracies and are better than the modified Burgers model. With increasing temperature, the accuracy of the three models all shows a decreasing trend. Asphalt mixtures are unstable at high temperatures, accelerating the transformation from elastic to viscous effects [42,43,44]. Therefore, in finite element analysis, the time hardening model has often been used to analyze the creep process of the material because it has a higher accuracy and fewer parameters, and it more easily converges in the iterative calculation process.

In the modified Burgers model, the elastic and viscoelastic deformation of the material during creep gradually decreases, and the viscous deformation gradually increases as the temperature increases [42]. Therefore, the elastic moduli *E*_1_ and *E*_2_ decrease, while the viscosity coefficient *η*_2_ also decreases. In the time hardening model, the increase in temperature enhances the creep properties of the mixture, and the creep rate increases, leading to a gradual rise in parameter *C*_1_. Similarly, in the modified time hardening model, the parameter *D*, representing the permanent deformation characteristics of the asphalt mixtures, also increases with temperature. According to Equations (19) and (20), the parameters n and m contain the same functional relationship as the parameters *C*_2_ and *C*_3_. First, the relationship between *n* and *C*_2_ is obtained by −*k*/*r*, the relationship between *C*_3_ and m are verified separately. The validation results are shown in Table 7.

The fitting accuracy affects the relationship between the two parameters. At low temperatures, the predicted values are close to the actual values. The relationship between the two parameters is more significant. When the temperature increases, the accuracy of the model fit decreases, and the relationship between the parameters *n*, *m*, *C*_2_, and *C*_3_ is not apparent. Therefore, the theoretical establishment process of the modified time hardening model is valid. In addition, the modified time hardening model, like the modified Burgers model, can represent the consolidation effect in creep. When time t tends towards infinity, both models show that strain *ε* tends to be stable. The creep limits obtained for the two models at different temperatures and stresses can be derived from Table 8 and Table 9. It is reasonable that the strain limit increases with increasing temperature and stress. Compared with the results obtained by the modified Burgers model, the predictions of both models are almost equal, which indicates that both models can accurately simulate the limiting values of viscoelastic deformation of asphalt mixtures.

## 6. Conclusions

This paper develops a model to represent the consolidation effect in the viscoelastic deformation of asphalt mixtures. The single penetration creep tests of a high-elasticity modified asphalt mixture were carried out under three temperatures (20 °C, 40 °C, and 60 °C) and four loads (0.55 MPa, 0.7 MPa, 0.85 MPa, and 1.00 MPa). The test results are simulated with the modified Burgers model, the time hardening model, and the modified time hardening model. Some crucial points can be made based on this study as follows:The study of the power law model considering the influence factor is successful, and this method allows the translation of the standard formula into the existing empirical model.Time and temperature have a greater effect on the creep process of asphalt mixtures, where the effect of stress is greater than that of temperature.The logistic model is a power function model, and the correction of the creep strain and creep compliance in this way does not affect the accuracy of the model in representing the creep process. The modified time hardening model can show the consolidation effect of the viscoelastic deformation of the asphalt mixture.Empirical models are an effective way to express creep, and such models are often more accurate than mechanical models in terms of fitting. The proposed modified time hardening model is comparable to the modified Burgers model to predict the deformation extremes. In addition, when the temperature is higher, the accuracy of fitting the three models is lower.Temperature is the main factor that affects the fitting precision of the three models. The higher the temperature, the lower the fitting accuracy of the model, and the less noticeable the consolidation effect of creep.The model presented in this paper is only validated for high-elasticity modified asphalt mixtures. This model is likely to be applied in studying the viscoelasticity of other materials in the future.

## Figures and Tables

**Figure 1 materials-15-02710-f001:**
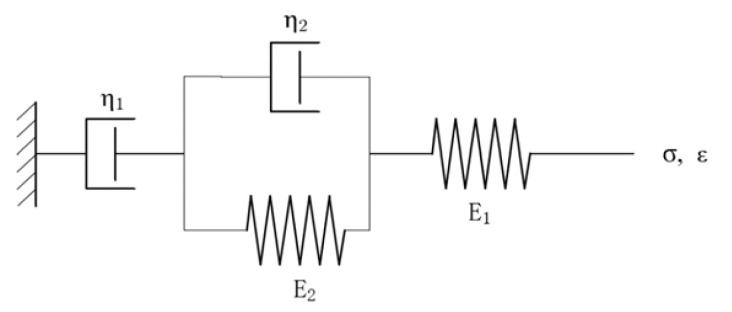
Burgers model with four units and four parameters.

**Figure 2 materials-15-02710-f002:**
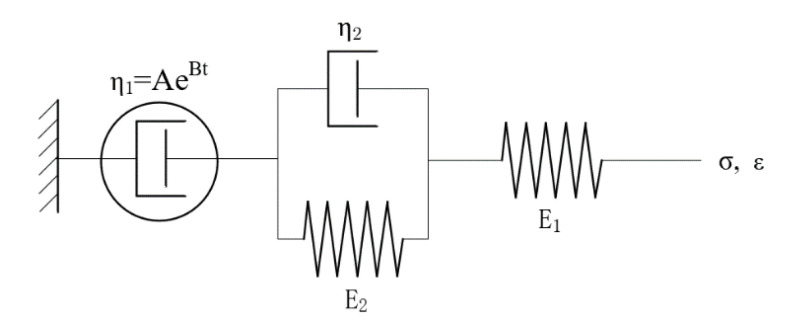
Burgers model with four units and five parameters.

**Figure 3 materials-15-02710-f003:**
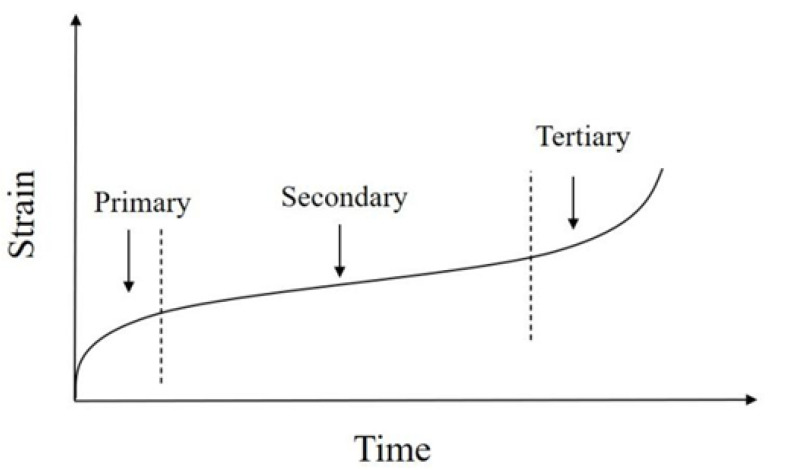
Three stages of creep behavior.

**Figure 4 materials-15-02710-f004:**
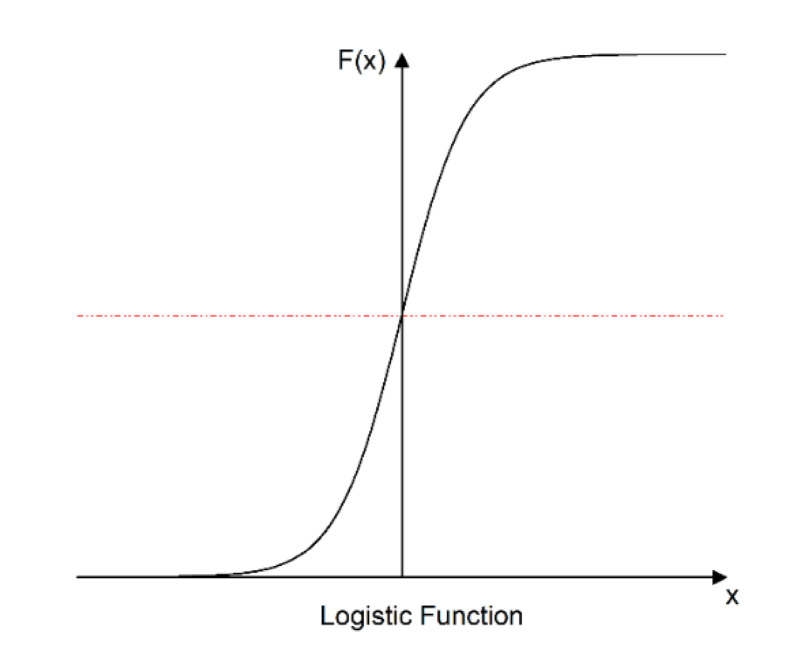
A typical trend of Logistic function.

**Figure 5 materials-15-02710-f005:**
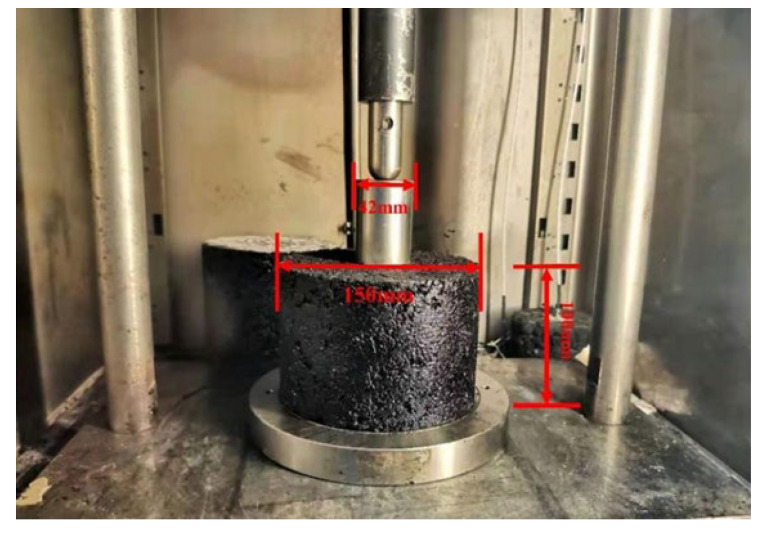
Loading device for uniaxial penetration creep test.

**Figure 6 materials-15-02710-f006:**
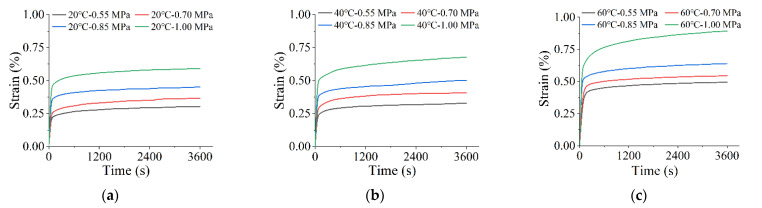
Effect of stress on time–strain creep curves at the same temperature level: (**a**) 20 °C; (**b**) 40 °C; (**c**) 60 °C.

**Figure 7 materials-15-02710-f007:**
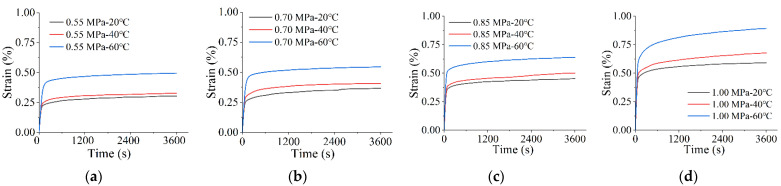
Effect of temperature on time–strain creep curves at the same stress level: (**a**) 0.55 MPa; (**b**) 0.70 MPa; (**c**) 0.85 MPa; (**d**) 1.00 MPa.

**Figure 8 materials-15-02710-f008:**
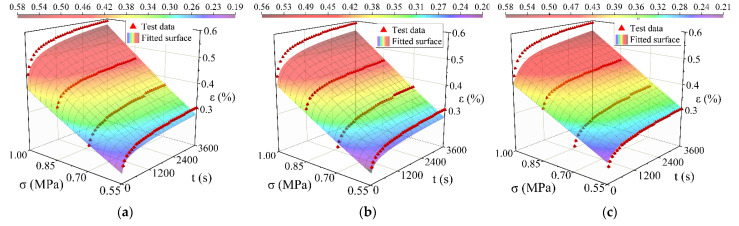
Comparison between test data points with the different model fitted surfaces at 20 °C: (**a**) Fitting result of the time hardening model; (**b**) Fitting result of the modified time hardening model; (**c**) Fitting result of the modified Burgers model.

**Figure 9 materials-15-02710-f009:**
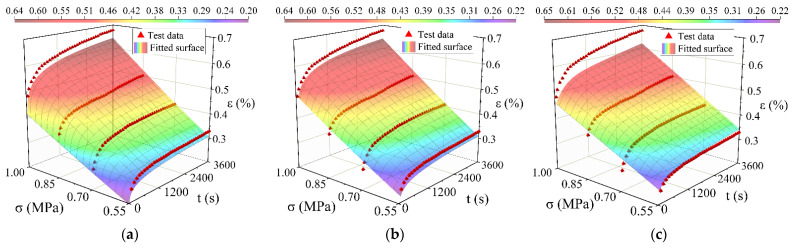
Comparison between test data points with the different model fitted surfaces at 40 °C: (**a**) Fitting result of the time hardening model; (**b**) Fitting result of the modified time hardening model; (**c**) Fitting result of the modified Burgers model.

**Figure 10 materials-15-02710-f010:**
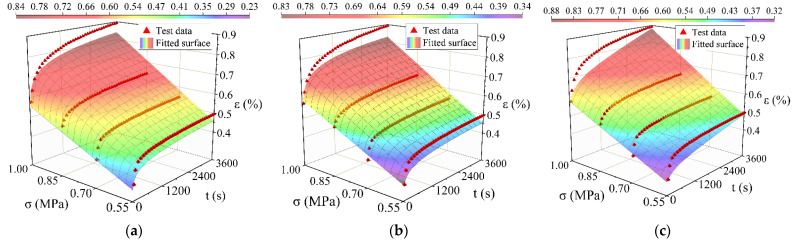
Comparison between test data points with the different model fitted surfaces at 60 °C: (**a**) Fitting result of the time hardening model; (**b**) Fitting result of the modified time hardening model; (**c**) Fitting result of the modified Burgers model.

**Table 1 materials-15-02710-t001:** Properties of the high-elasticity modified asphalt binder.

Test	Unit	Specification	Results	Test Methods
Softening point	°C	>60	87.5	T 0606-2011
Pen (25 °C)	0.1 mm	60~100	84	A041
Ductility (10 °C)	cm	≥50	81	A043
toughness (25 °C)	N ∙ m	≥12	25	A057
Flashpoint	°C	≥280	348	A045
Specific gravity (15 °C)	g/cm^3^	1.09	≥1.00	A049
Ash	%	0.07	≤1.00	T0614-2011

**Table 2 materials-15-02710-t002:** The strain rate of increases at different stress intervals (for 0.55 MPa).

Stress Intervals	20 °C		40 °C		60 °C	
	SROI	STD	SROI	STD	SROI	STD
0.55 MPa~0.70 MPa	18.9%	1.71%	24.9%	1.32%	10.9%	1.78%
0.55 MPa~0.85 MPa	52.0%	3.95%	51.2%	2.96%	31.0%	13.30%
0.55 MPa~1.0 MPa	99.4%	3.86%	103.9%	3.00%	76.8%	12.10%

**Table 3 materials-15-02710-t003:** The relative strain growth rate at different temperature intervals (for 20 °C).

Temperature Intervals	0.55 MPa		0.70 MPa		0.85 MPa		1.00 MPa	
	SROI	STD	SROI	STD	SROI	STD	SROI	STD
20 °C~40 °C	8.91%	1.27%	14.40%	2.08%	8.36%	1.68%	11.40%	2%
20 °C~60 °C	65.40%	9.48%	54.10%	8.41%	41.80%	5.80%	46.20%	4.68%

**Table 4 materials-15-02710-t004:** Fitting parameters of the time hardening model.

Temperature	C_1_	C_2_	C_3_	Regression Indicator	
				R^2^	RMSE
20 °C	9.18 × 10^−12^	1.231	−0.933	0.978	1.87 × 10^−4^
40 °C	9.23 × 10^−12^	1.242	−0.925	0.976	2.15 × 10^−4^
60 °C	3.91 × 10^−9^	1.014	−0.93	0.932	4.64 × 10^−4^

**Table 5 materials-15-02710-t005:** Fitting parameters of the modified time hardening model.

Temperature	D	a	n	m	Regression Indicator	
					R^2^	RMSE
20 °C	1.03 × 10^−8^	1030.26	−0.438	−0.13	0.978	1.87 × 10^−4^
40 °C	1.27 × 10^−8^	1060.21	−0.426	−0.135	0.976	2.16 × 10^−4^
60 °C	1.43 × 10^−8^	3.19	−0.020	−0.148	0.932	4.65 × 10^−4^

**Table 6 materials-15-02710-t006:** Fitting parameters of the modified Burgers model.

Temperature	E_1_ (MPa)	A (×10^5^ MPa)	B (×10^−3^ s^−1^)	E_2_ (MPa)	η_2_ (×10^12^)	Regression Indicator	
						R^2^	RMSE
20 °C	241.8	4.118	2.66	189.0	6.34	0.961	2.64 × 10^−4^
40 °C	225.7	3.561	2.91	137.3	4.08	0.953	3.85 × 10^−4^
60 °C	173.0	3.254	3.78	129.8	1.37	0.920	5.02 × 10^−4^

**Table 7 materials-15-02710-t007:** Verification analysis of parameters *m* and *C*_3_.

Temperature	$-\frac{k}{r}\;$	$-\frac{{k(C}_{3}+1)}{r}\;$	m	R^2^
20 °C	−1.896	−0.127	−0.130	0.978
40 °C	−1.760	−0.132	−0.135	0.976
60 °C	−1.428	−0.100	−0.148	0.932

**Table 8 materials-15-02710-t008:** Prediction of limit strain by the modified Burgos model.

Temperature	$\varepsilon =\sigma (\frac{1}{E_{1}}+\frac{1}{AB}+\frac{1}{E_{2}})\;$			
	*σ* = 0.55 MPa	*σ* = 0.70 MPa	*σ* = 0.85 MPa	*σ* = 1.00 MPa
20 °C	0.005687	0.007238	0.008789	0.010340
40 °C	0.006973	0.008875	0.010777	0.012679
60 °C	0.007864	0.010008	0.012153	0.014298

**Table 9 materials-15-02710-t009:** Prediction of limit strain by the modified time hardening model.

Temperature	ε=σ·D			
	*σ* = 0.55 MPa	*σ* = 0.70 MPa	*σ* = 0.85 MPa	*σ* = 1.00 MPa
20 °C	0.005687	0.007238	0.008789	0.01034
40 °C	0.006974	0.008876	0.010778	0.01268
60 °C	0.007865	0.01001	0.012155	0.01430

## Data Availability

The data presented in this study are available on request from the corresponding author.

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
