# Peer review of "Study of a Modified Time Hardening Model for the Creep Consolidation Effect of Asphalt Mixtures"

_materials, 2022, doi:10.3390/ma15082710_

Round 1

Reviewer 1 Report

The paper title “Study of a modified time-hardening model for the creep consolidation effect of asphalt mixtures” by Ma et al. describes the consolidation effect of an asphalt mixture during the viscoelastic deformation process through establishing a modified time-hardening model based on the expression of the existing power function model. The manuscript needs major revision before consideration to be published in materials journal for the following reason:

  • The manuscript was not arranged according to the journal template, also, the style of reference was not matched with the journal style.
  • The hypothesis of the current study should be clarified at the end of the introduction
  • The title of the figures should be mentioned in more detail.
  • The symbols in equations should be clarified after each equation, please check throughout the manuscript
  • The abbreviation should be mentioned complete at the first time and mentioned as abbreviate after that such as Chinese national standards (CNS), it should be mentioned like this in the first time and CNS after that, please revise this throughout the manuscript
  • Figure 7 can be comprised in one figure containing more panels, this is also for Fig. 8.
  • Please add a title for Figure 8
  • Authors should be following the common structure for paper, introduction, Material and Methods, Results, Discussion, and Conclusion to avoid conflict during reading.
  • Authors should be revised the English language throughout the manuscript carefully

Author Response

Thank you for your comments. The manuscripts have been arranged in accordance with the requirements of the journal and the format of the reference has been completely revised. Please see the attachment for details.

Reviewer 2 Report

In this article, the authors discussed the model regarding asphalt mixtures. However, there are few issues that need to be taken care of during revised manuscript as follows:

  1. What is the effect of diameter and height of the material on its performance? A valid explanation is required.
  2. Does the material shows the same trend for higher temperature and pressure range? This should be added with proof in the revised manuscript.
  3. It has been mentioned that “The whole process of creep is affected by time t, stress q, and temperature T”. However the effect of time has not been properly explained? Proper explanation is required.
  4. What is the significance of Table 2? It should be rectified.
  5. The authors should use same writing style for all the references.

Author Response

Thank you for your comments. The size of the specimen has different influence on the mechanical response and penetration depth in creep test, but it does not change the viscoelastic properties of the material. Therefore, an explanation is added in the article "2.3 Specimen preparation": considering the influence of Specimen radius and height on creep behavior, the Specimen prepared according to the standard AASHTO T 312 is the standard size, the size effect is eliminated.  Please see the attachment for details.

Reviewer 3 Report

This paper is technically interesting; however, it needs some important modifications before it can be accepted for publication. The authors are asked to address the following comments:

  1. The novelty of this paper should be further justified and to establish the contributions to the new body of knowledge.
  2. In Introduction section, the authors should improve the research background, the review of significant works in the specific study area, and the knowledge gap.
  3. The literature review should be extended to more recently published works available in the literature. Some of the previous studies that investigated similar topics are listed below:

(a) Effect of super absorbent polymer on microstructural and mechanical properties of concrete blends using granite pulver, Structural Concrete 22, E898-E915.

  1. The conclusion section needs improvement, more key findings and how the findings would help the researchers and industry needs to be stated.
  2. There are some grammatical mistakes and typos, which needs to be correct. A thorough proofread would be appreciated.
  3. Abstract section should be improved considering the following structure: introduction, problem statement, methodology, results, and conclusion.
  4. The presentation of the results and conclusions could be improved.

Author Response

Thank you for your comments. 

The research background of the abstract and the introduction has been detailed, and the innovation of the article has been highlighted.

Please see the attachment for details

Reviewer 4 Report

The paper presents an experimental study concerning the results of a single penetration creep test on modified high elasticity asphalt mixtures at different temperatures (20℃, 40℃, 60℃) and loading conditions (0.55 MPa, 0.70 MPa, 0.85 MPa, 1.00 MPa). The authors use the results for a simulation by fitting a model to the experimental data obtained.

This reviewer has some considerations for the authors, as follows.
The paper is unformatted according to the journal's template. The formatting and quality of the images presented are poor. Authors need to improve this foundation.

There are several points in the text with writing, typing, and formatting errors. Do a full review.

The introduction needs to be improved by further discussing the research problem. Also include authors/studies from other continents.

Chapter 2 is very long (almost a third of the paper) and fragmented.
The materials and methods chapter needs to be improved. It is very superficial, with unnecessary images and tables.

Table 2 and Figure 5 are not required.

Improve the results chapter. Join images a), b), and C) of figures 7 and 8, leaving them all on the same y-axis scale. Improve the quality of graphics presentation. Tables 3 and 4 present the standard deviations obtained.

In figures 9 to 11, it is impossible to read the information.

No statistical information about modeling errors was presented.

A question that needs to be answered by the authors concerns the model adjustment with the experimental study results. Logically, the model after adjustment will reduce the error for the database used. The question that arises is can the fitted model represents other similar databases?

The authors state, "This model improves the original empirical model" in the conclusions. However, as indicated above, this cannot be generalized as the model was only improved with the study database.

Author Response

Thank you for your comments.

We have revised it according to the format requirements of the journal. And uploaded some images to reproduce.

Please see the attachment for details.

Round 2

Reviewer 1 Report

The manuscript should be rearranged in accordance with the requirements of the journal (please use the word journal template). Also, the title of the figures needs more clarifications.  

Author Response

Thank you for your comments.  The manuscript has been rearranged in accordance with the requirements of the journal and the title of some figures has been revised. 

Reviewer 2 Report

NA

Author Response

Thank you for your comments.

Reviewer 3 Report

Thanks for addressing my comments. The paper can now be accepted for publication. 

Author Response

Thank you for your comments.

Reviewer 4 Report

Dear authors

The authors made most of the suggested corrections, which improved the overall quality of the paper.

However, minor revisions are still needed:
The template used is not correct. Please see in
https://susy.mdpi.com/user/manuscripts/upload/3e702058002de96590e6ec19c80d3815?form%5Bjournal_id%5D=14

The introduction needs to be improved. It still didn't fit. Better not insert an image in the introduction.

The figures still need to be improved.
Is figure 6 indispensable?
Figures 8 should be grouped and presented side by side or below the other, like a graphic board. Same for figure 9.
Figures 10, 11, and 12 are not legible. The information is all distorted. What software are the authors using to produce the images? I believe he is not suitable.

Author Response

Thank you for your comments. 

The manuscript has been arranged in accordance with the requirements of the journal. The introduction has been improved and some figures have been revised.